# Immediate Effects of Dry Needling on the Autonomic Nervous System and Mechanical Hyperalgesia: A Randomized Controlled Trial

**DOI:** 10.3390/ijerph18116018

**Published:** 2021-06-03

**Authors:** Irene Lázaro-Navas, Cristina Lorenzo-Sánchez-Aguilera, Daniel Pecos-Martín, Jose Jesús Jiménez-Rejano, Marcos Jose Navarro-Santana, Josué Fernández-Carnero, Tomás Gallego-Izquierdo

**Affiliations:** 1Department of Physical Therapy, University Hospital Ramón y Cajal, 28034 Madrid, Spain; ireneLN88@gmail.com; 2University of Alcalá, Instituto de Fisioterapia y Dolor, 28805 Madrid, Spain; cris_lorenzo85@hotmail.com (C.L.-S.-A.); daniel.pecos@uah.es (D.P.-M.); tomas.gallego@uah.es (T.G.-I.); 3Department of Physical Therapy, University of Alcalá, 28805 Alcalá de Henares, Spain; 4Department of Physical Therapy, Faculty of Nursing, Physiotherapy and Podology, University of Sevilla, 41009 Sevilla, Spain; jjjimenez@us.es; 5Rehabilitation San Fernando, Alcalá de Henares, 28801 Madrid, Spain; marcosjose.navarrosantana@gmail.com; 6Department of Physical Therapy, Occupational Therapy, Rehabilitation and Physical Medicine, Rey Juan Carlos University, 28922 Madrid, Spain; 7Grupo Multidisciplinar de Investigación y Tratamiento del Dolor, Grupo de Excelencia Investigadora URJC-Banco de Santander, Universidad Rey Juan Carlos, 28032 Madrid, Spain; 8Paz Hospital Institute for Health Research, IdiPAZ, 28029 Madrid, Spain; 9Motion in Brains Research Group, Institute of Neuroscience and Sciences of the Movement (INCIMOV), Centro Superior de Estudios Universitarios La Salle, Universidad Autónoma de Madrid, 28023 Madrid, Spain

**Keywords:** dry needling, autonomic nervous system, physiological effects, cortisol, pain physiology

## Abstract

Background: Dry needling (DN) is often used for the treatment of muscle pain among physiotherapists. However, little is known about the mechanisms of action by which its effects are generated. The aim of this randomized controlled trial was to determine if the use of DN in healthy subjects activates the sympathetic nervous system, thus resulting in a decrease in pain caused by stress. Methods: Sixty-five healthy volunteer subjects were recruited from the University of Alcala, Madrid, Spain, with an age of 27.78 (SD = 8.41) years. The participants were randomly assigned to participate in a group with deep DN in the adductor pollicis muscle or a placebo needling group. The autonomic nervous system was evaluated, in addition to local and remote mechanical hyperalgesia. Results: In a comparison of the moment at which the needling intervention was carried out with the baseline, the heart rate of the dry needling group significantly increased by 20.60% (SE = 2.88), whereas that of the placebo group increased by 5.33% (SE = 2.32) (*p* = 0.001, d = 1.02). The pressure pain threshold showed significant differences between both groups, being significantly higher in the needling group (adductor muscle *p* = 0.001; d = 0.85; anterior tibialis muscle *p* = 0.022, d = 0.58). Conclusions: This work appears to indicate that dry needling produces an immediate activation in the sympathetic nervous system, improving local and distant mechanical hyperalgesia.

## 1. Introduction

Dry needling (DN) can be defined as “a technique in which a fine needle is used to penetrate the skin, subcutaneous tissues, and muscle, with the intent to mechanically disrupt tissue without the use of an anesthetic” [1]. DN is a safe and minimally invasive technique [2] typically used for the treatment of an assortment of neuromusculoskeletal pain syndromes [3,4,5]. In recent decades, DN has been widely used to treat muscle pain and, more specifically, the treatment of the myofascial trigger point (MTrP). The MTrP is a “hyperirritable spot within a taut band of skeletal muscle that is painful on compression, stretch, overload, or contraction of the tissue which usually responds with a referred pain that is perceived distant from the spot” [6]. MTrPs are classified as being active (in the event of spontaneous pain) or latent (absence of spontaneous pain) [7]. The presence of MTrPs in the skeletal muscle has been associated with an impaired range of motion, muscle weakness, loss of coordination, pain, and autonomic reactions [7,8]. DN may have been shown to be effective in the management of pain, improving range of motion, muscle strength, and coordination [9]. The possible explanations found in the literature for the decrease in pain include the effects of dry needling at the local level (producing an interruption of spontaneous electrical activity on the taut band or local vasodilation), activation of the peripheral segmental pain inhibition (explained through Gate Control Theory), or activation of the descending pathways of pain inhibition at the central nervous system level (serotonergic and noradrenergic endogenous opioid release and conditioned modulation of pain) [8]. However, these underlying mechanisms have not yet been fully clarified and are not understood [1,9].

Another possible DN mechanism for the modulation of pain is stress-induced analgesia (SIA) [10]. SIA has been described as “a reduced nociceptive response after stress exposure, which is mediated by descending inhibitory opioid and nonopioid brain circuits” [11]. SIA is influenced by activity of the autonomous nervous system (ANS) and the hypothalamic–pituitary–adrenal (HPA) axis [12,13]. The ANS, prior to a stressful stimulus, quickly induces physiological changes through synaptic transmissions via two branches, the sympathetic and parasympathetic nervous systems, resulting in an increase in sympathetic nervous system (SNS) activity [13]. SNS activity is usually determined by measuring skin conductance [14], heart rate, and respiratory rate values [15]. Other therapeutic procedures have been shown to produce sympathoexcitatory changes that had therapeutic benefits for patients [16]. HPA axis activity is measured by determining the cortisol level in saliva. Cortisol is an anti-inflammatory hormone regulated by the HPA axis via feedforward and feedback loops, which is related to the modulation of nociception and stress-induced analgesia [17]. It appears that the cortisol level in saliva increases with a stressful event which, in turn, appears to be related to the SIA process [18]. Therefore, the activation of the neuroendocrine system SNS–HPA axis maintains homeostasis and produces an analgesic effect [19].

Given the nature of DN [20], which is a technique that can be considered stressful, SNS–HPA axis activity could be among the possible physiological mechanisms that explain its analgesic effect [9]. Nevertheless, to our knowledge, no research has been undertaken that explores whether the SNS–HPA axis is involved in the response to DN.

The purpose of this study was to determine if the application of a DN technique results in changes in skin conductance, heart rate, temperature, breathing rate, or cortisol levels in saliva between different measurements, in addition to assessing improvements in the pressure pain threshold. We hypothesized that a DN technique would result in an activation of the SNS and HPA axis, which plays a crucial role in pain modulation.

## 2. Materials and Methods

### 2.1. Study Design

A randomized controlled clinical trial of parallel groups was carried out to compare a deep DN treatment with a placebo treatment, with the aim of evaluating the effects produced on the ANS and on pain. The study was performed following the CONSORT 2010 (Consolidated Standards of Reporting Trials) [21,22] directives and the STRICTA (Revised Standards for Reporting Interventions in Clinical Trials of Acupuncture) [23] criteria. It was approved by the Committee of Research Ethics and Animal Experimentation of the University of Alcalá (CEIT/HU/2015/06 of 23 November 2015) and registered in the Australian New Zealand Clinical Trials Registry at http://www.anzctr.org.au/ (ACTRN12616000881437) (accessed on 29 May 2021).

### 2.2. Participants

The sample comprised healthy volunteers from the student body of different degrees and the administrative staff of the University of Alcalá. These were selected using convenience non-probability sampling.

The following inclusion criteria were established: (1) 18 to 65 years of age, (2) pain-free, and (3) latent MTrP in the adductor muscle of the left thumb. Participants were excluded if: (1) acute illness was present at the time of the study; (2) any of the following were present: fibromyalgia, diabetes, cardiopathy, essential arterial hypertension, hemophilia, a neurological disease, cognitive decline, a compromised immune system (HIV, cancer, hepatitis, acute immune diseases), left upper limb lymphedema; (3) taking blood thinners; (4) pregnancy; (5) fear of needles; (6) allergic to metals (nickel or chrome); (7) had participated in a dry needling/acupuncture study in the past 6 months.

The subjects who met the selection criteria were informed of the study procedure by an information sheet, and they signed a data release document and provided informed consent, according to the standards of the Declaration of Helsinki.

### 2.3. Randomization and Blinding

Each participant was assigned a code. The subjects were also randomly assigned to receive a deep DN technique or a placebo needling technique. Those not included in the study performed a concealed randomization with the program Epidat 4.2, using a 1:1 allocation ratio through simple randomization.

Both the examiner who obtained all of the outcome measures and the statistician who examined the data were blinded to the subject randomization. The subjects participating in the study were blinded to the intervention, because they were not informed about the existence of a placebo group. The physiotherapist assessed the MTrPs before knowing the intervention group to which each subject belonged.

### 2.4. Procedure

Interventions were carried out at the School of Nursing and Physiotherapy of the University of Alcalá (Madrid, Spain). All of the subjects received one session of deep dry needling or placebo needling. Measurements were performed between 9:00 a.m. and 11:00 a.m. [24]. The temperature of the room was maintained in the range of 24–25 °C, and the noise level was kept to a minimum. The subjects could neither ingest alcohol or caffeine, nor perform vigorous physical activity, on the day of the study. They were not allowed to smoke during the 2 h before the study. Moreover, they could not brush their teeth, ingest any liquids, eat solid food, or chew gum 30 min before the study.

The subjects lay supine on a stretcher, with their forearms free and legs stretched out. A professional physiotherapist with experience (about 15 years) in the palpation, diagnosis, and treatment of MTrPs and myofascial pain syndrome was responsible for locating and marking the MTrPs. Thus, he was considered a qualified examiner, with a good reproducibility index (k = 0.63) [25]. Subjects were instructed to remain calm and quiet, but completely awake. A ten-minute period was predetermined for the subject to acclimate to the room conditions before beginning the recording of the physiological variables [26,27].

The physiotherapist carried out the interventions in both groups, on a latent MTrP in the adductor muscle of the left-hand thumb. The same aseptic measures pertinent to the technique were applied for all subjects in both groups. Due to the supine position on the stretcher, the vision of the subjects of the DN technique was blocked. The procedure followed in the two groups is described below.

#### 2.4.1. Deep Dry Needling Group

Deep DN was performed with disposable needles (0.25 × 0.25 mm; AGU-A1038P; Agu-Punt S.L, Barcelona, Spain) [28]. The “fast-in and fast-out” technique asserted by Hong [29,30] was employed. The needle was moved up and down in multiple directions (vertically, without rotations), at approximately 1 Hz frequency for 10 s, to look for LTRs [28].

#### 2.4.2. Placebo Needling Group

A non-penetrating, simulated DN technique with placebo needles was applied to the subjects who were randomized into this group, with a modification of the protocol developed by Tough et al. [31]. Disposable sterile needles (0.25 × 0.40 mm; DB100-2540; DongBand, AcuPrime^®^, Exeter, UK) were used. These needles have a red tab and cannot be distinguished from those used in the Deep Dry Needling Group (Figure 1). They also have a spring handle that can be glided up and down, imitating the movement in and out of the skin, without genuine penetration. To achieve an effective blinding [32], all of the needles were held in the box used for the Deep Dry Needling Group. The professional pinched the MTrP of the adductor muscle of the thumb, placed the guide-tube with the needle exerting some pressure, and jabbed the needle against the skin, simulating an insertion. Next, he withdrew the guide-tube, pressing the tip of the needle with his thumb, making sure the needle did not move (to ensure it did not penetrate the skin). The needle stayed in contact with the skin each time. Then, he moved the spring handle up and down 10 times at a speed of 1 Hz, in a “sparrow pecking” movement. Each time the handle moved up and down, the pressure sensation increased, replicating the feeling of a puncture.

### 2.5. Outcome Measures

#### 2.5.1. Psychological Factors

Before beginning the intervention, participants completed different psychometric tests to assess their initial levels of depression (the Beck Depression Inventory II (BDI-II)) [33], anxiety (the State-Trait Anxiety Inventory (STAI)) [34], and pain catastrophizing (Pain Catastrophizing Scale (PCS)) [35]. This testing was undertaken because high levels in these variables can affect ANS activity and HPA activity.

#### 2.5.2. Autonomic Nervous System Assessments: Primary Outcomes

##### Physiological Variables

To measure the ANS response when applying a DN technique, multiparametric biofeedback equipment NeXus 10 MK-II was used (Mind Media BV; Herten, the Netherlands) [36,37,38]. Data were processed using the Biotrace software, version V2015B (Mind Media BV). The skin conductance (SC) and peripheral temperature of the skin (Temp) were registered with the sensors placed as shown in Figure 2; heart rate (HR) was registered with EKG sensors, and breathing rate (BR) with a sensor placed on the sternum with an elastic band.

Measurements for the physiological variables were collected at different moments (Figure 3): Baseline (average of the 5 min before the intervention); Dry needling (average of the 10 s of the needling technique); Post-1 (average of 1 min immediately after ending the intervention); Post-2 (average of 1 min, 9 min after ending the intervention).

##### Cortisol

Free cortisol levels in saliva were determined using the Cortisol ELISA^®^ kit from IBL International laboratories (Hamburg, Germany) [39,40]. Compared to cortisol determination in plasma, this is a simple, painless, and non-invasive method. It is also less costly, no specialized medical staffing is required, and does not produce stress when performing the vein puncture [41,42]. In addition, the correlation coefficient between the two methods is r > 0.9 [39]. The saliva samples were collected before the intervention and three minutes after it ended [24] (Figure 3). Cortisol levels in the saliva were analyzed in the laboratories of the Research Foundation of the University Hospital Príncipe de Asturias in Alcalá de Henares.

#### 2.5.3. Pain Assessments: Secondary Outcomes

##### Pain Intensity

The Numeric Rating Scale for Pain (NRS) was used to measure the subjective pain perceived by the participant during the needling technique performance. This is a valid and reliable measuring tool to assess pain intensity during a treatment and/or intervention [43], and presents a good correlation with the Visual Analog Scale (VAS), with an overall intraclass correlation coefficient (ICC) of >0.7 [44]. To implement this scale, the subjects were asked after the intervention (to avoid talking during the recording of the physiological measures) about the maximum pain experienced during the performed needling technique, selecting a whole number between 0 and 10 that best reflected their pain intensity (0: no pain and 10: worst imaginable pain).

##### Pressure Pain Threshold (PPT)

A Wagner Force Dial™ model FDK 20 (Greenwich, CT, USA) manual algometer was used to measure PPT before and after performing the intervention (Figure 3) on the two points previously marked: on the latent MTrP of the adductor muscle of the left thumb (point selected to examine the implicated area) and on the most painful point on palpation of the anterior tibialis muscle, approximately at 2.5 cm lateral and at 5 cm distal to the anterior tibial tuberosity [45] (point selected as an unrelated segment area to assess descending pain inhibitory mechanisms). For this purpose, the tool was placed perpendicular to the point previously marked, and through an approximate gradual increase in speed of 1 kg/s, pressure was increased until the subject started feeling pain or discomfort (without ever reaching the maximum bearable pressure); at that point, pressure was stopped. Three measurements were taken on the same point, with a 30-s time interval between each measurement, and the average of the three measurements was analyzed [16,28,46]. This method presents high reliability (ICC = 0.91) [46].

### 2.6. Sample Size

A prior pilot study was conducted with 20 subjects: 10 in the experimental group and 10 in the control group. In this research, the main variable was skin conductance (SC), measured with the biofeedback Nexus 10 MK-II equipment. A repeated measures contrast was used. The size effect was 0.143; a 0.05 alpha level and a power of 0.95% were assumed, plus 15% possible loss. These assumptions generated a simple size of 65 participants in total. The statistical analysis program G*Power 3.1.9.4 was used.

### 2.7. Statistical Analysis

Data were analyzed with the statistical package SPSS for Windows, version 26.0 (SPSS Science, Chicago, IL, USA). To study the homogeneity of the groups at baseline, Student’s *t*-test was used for independent samples in the quantitative variables, and the Pearson’s chi-Squared test was used for the qualitative variables.

To normalize the differences between participants in the variables analyzed, the data of each time period were evaluated in terms of percentage change (% Change), using the formula employed by Perry and Green [47].

Regarding the primary outcomes of the physiological variables, a separate 2-by-4 mixed model analysis of the variance was employed to assess the effects of the intervention, for which group (deep dry needling or control) was the between-subjects variable, and time (the different measurements) the within-subjects variable. An a priori alpha level of 0.05 was set. The hypothesis of interest was the group-by-time interaction. In addition, the effect size was estimated by calculating the partial Eta2 coefficient (ηp2). The difference between the two groups in the percentage change in all measurements was compared using Student’s *t*-test for independent samples, or, alternatively, Welch’s *t*-test. Bonferroni type adjustment of significance was used.

For the analysis of cortisol and PPT, a separate 2-by-2 mixed model analysis of variance was employed to assess the effects of the intervention, for which group (deep dry needling or control) was the between-subjects variable, and time (the different measurements) the within-subjects variable. An a priori alpha level of 0.05 was set. The hypothesis of interest was the group-by-time interaction. Effect size was estimated by calculating ηp2. The difference between both groups in % Change at baseline and at post-test was compared using Student’s *t*-test for independent samples, or, alternatively, Welch’s *t*-test. In addition, effect size was estimated using Cohen’s d, considering “small effect size” to be between 0.2 to 0.5, “medium effect size” 0.5 to 0.8, and “large effect size” greater than 0.8. In all statistical tests, the level of significance was set at 95% (*p* < 0.05; two-tailed test).

Finally, different correlations using Pearson’s correlation coefficient were studied. First, the values obtained from the Numeric Rating Scale for Pain (NRS) for each group are shown, in addition to the existing correlations in this scale regarding % Change between the baseline–dry needling measurement of the physiological variables and regarding % Change of cortisol. In addition, correlations between % Change of the Physiological Variables and Cortisol are shown in the different measures with the Psychological Factors.

## 3. Results

### 3.1. Participant Characteristics

A total of 65 subjects, with an average age of 27.78 years (SD ± 8.41 years), of whom 33 (50.8%) were men and 32 (49.2%) were women, met the selection criteria and participated in the study between November 2016 and February 2017. All of the data were collected for analysis, as shown in the flow diagram in Figure 4. Table 1 and Table 2 contain information about the characteristics at baseline of all of the subjects in each treatment group, confirming no statistically significant differences between the groups. It can also be observed in Table 1 that the subjects did not present signs of depression, anxiety, or pain catastrophizing. Table 2 also shows the values obtained regarding pain intensity perceived during the intervention. The NRS results were significantly higher in DN group than in the placebo group.

### 3.2. Autonomic Nervous System Assessments: Primary Outcomes

The mixed model analysis of the variance of the variable HR indicated a statistically significant group-by-time interaction (F = 9.99, *p* < 0.001, ηp2 = 0.137), and a time effect (Dry Needling Group F = 50.53, *p* < 0.001, ηp2 = 0.612; Placebo Group F = 12.57, *p* < 0.001, ηp2 = 0.288). In this variable, no effect was found in the group factor (F = 0.04, *p* = 0.845, ηp2 = 0.001). As shown in Table 3, comparing the Baseline with the Needling, the % Change of the HR significantly increased in the dry needling group compared to the placebo group. Subsequently, the % Change decreased more in the placebo group at Post-1, whereas the % Change from Post-1 to Post-2 was significantly greater for the HR of the dry needling group.

In the variables SC, Temp, and BR, the mixed model analysis of variance indicates no statistically significant group-by-time interaction (SC F = 0.91, *p* = 0.380, ηp2 = 0.014; Temp F = 1.80, *p* = 0.173, ηp2 = 0.028; BR F = 0.52, *p* = 0.667, ηp2 = 0.008), but does indicate a time effect (SC: Dry Needling Group F = 71.88 *p* < 0.001, ηp2 = 0.692; Placebo Group F = 72.16, *p* < 0.001 ηp2 = 0.700/Temp: Dry Needling Group F = 3.52, *p* = 0.042, ηp2 = 0.099; Placebo Group F = 6.84, *p* = 0.003, ηp2 = 0.181/BR: Dry Needling Group F = 28.33, *p* < 0.001, ηp2 = 0.470; Placebo Group F = 25.64, *p* < 0.001, ηp2 = 0.453). No effect was found in the group factor (SC F = 2.08, *p* = 0.154, ηp2 = 0.032; Temp F = 0.75, *p* = 0.389, ηp2 = 0.012; BR: F = 2.50, *p* = 0.119, ηp2 = 0.038).

A considerable increase in SC was observed after needling compared to the baseline measurement, although no significant differences existed between the two groups. However, a greater decrease (*p* < 0.001) was produced in the SC values of the Placebo group (−21.50%, SE = 1.99) in comparison with the Dry Needling Group (−10.36%, SE = 1.96) in the Post-1 measurement after needling (Table 3). On the contrary, no differences were found in the variables BR and temperature between groups in the percentage change of any of the measurements performed (Table 3).

Regarding cortisol, no significant group-by-time interaction was found (F = 2.07, *p* = 0.155, ηp2 = 0.032), nor significant differences between groups (F = 0.002, *p* = 0.969, ηp2 < 0.001). Notwithstanding the data, an increase of 11.27% (SE = 4.76) was produced in the Dry Needling group, with a difference existing in the time factor (F = 6.84, *p* = 0.013, ηp2 = 0.176). On the contrary, cortisol increased only 1.51% (SE = 3.08) in the Placebo group, and no difference was recorded in the time factor (F = 0.64, *p* = 0.428, ηp2 = 0.02).

### 3.3. Pressure Pain Threshold

As shown in Table 2, the variance mixed model analysis indicated a significant interaction between the intervention group and the time (different measures) in the algometry thumb adductor (F = 77.88, *p* < 0.001, ηp2 = 0.553) and algometry of anterior tibialis muscle (F = 50.24, *p* < 0.001, ηp2 = 0.444), and a time effect (Algometry thumb adductor: Dry Needling Group F = 232.92, *p* < 0.001, ηp2 = 0.879; Placebo Group F = 58.06, *p* < 0.001, ηp2 = 0.652/Algometry of anterior tibialis muscle: Dry Needling F = 139.19, *p* < 0.001, ηp2 = 0.813; Placebo F = 47.54, *p* < 0.001, ηp2 = 0.605). However, no effect was found in the group factor (Algometry thumb adductor F = 2.23, *p* = 0.14,1 ηp2 = 0.034; Algometry of anterior tibialis muscle F = 0.84, *p* = 0.364, ηp2 = 0.013). Additionally, a statistically significant difference was observed in the percentage change (Table 3) between the measurement Baseline and the Post-test in PPT, both in the adductor of the thumb and in the anterior tibialis. In both cases, this percentage was higher in the Dry Needling group compared to the Placebo group, and the effect size was “large” (adductor of the thumb Cohen’s d = 1.87; anterior tibialis Cohen’s d = 1.61).

### 3.4. Correlations between Pain Intensity, Physiological Variables and Cortisol

Regarding the analyzed correlations between the Numeric Rating Scale for Pain, and the physiological variables and cortisol, no significant correlations were found (Table 4). The correlations between the physiological variables and cortisol, in addition to the Psychological Factors, are shown in Table 5 for the Dry Needling Group. A modest positive correlation (*p* = 0.046, r = 0.350) was found in % Change of the temperature between Needling and Post-1 with State Anxiety; and in the BR variable in the % Change Post-1 to Post-2 with the Total PCS (*p* = 0.024, r = 0.393). The results for the Placebo Group are shown in Table 6, and indicate a significant negative correlation for the BR in the % Change between Post-1 and Post-2 with State Anxiety (*p* = 0.047, r = −0.354), and between % Change Cortisol and the BDI-II (*p* = 0.021, r = −0.407), State Anxiety (*p* = 0.018, r = −0.416), and Trait Anxiety (*p* = 0.037, r = −0.371).

## 4. Discussion

The aim of this study was to evaluate the effects of DN on the ANS and nociceptive processing, by applying DN using the traditional method of fast-in and fast-out technique. The results of this study demonstrate that DN has a neurophysiological effect on the ANS and pain processing, showing an increased heart rate and an increased pressure pain threshold, both locally and at remote sites, compared to the placebo. However, only changes in heart rate were found, and no changes were found in the remainder of the measured parameters of the ANS. The heart rate undergoes innervation from the ANS but not exclusively [48]. These results are supported by a recently published study in which DN was evaluated on the cervical paravertebral muscles. It was found that patients experimented ANS changes, which were detected by measuring pupillometry [49]. In addition, this research found that leaving the needles in the muscle for 21 min caused an activation of the ANS that lasted 18 min, before returning to the basal state.

These outcomes are consistent with those of several previous studies [50,51]. Haker et al. [50] investigated the effects of acupuncture applied over the thenar muscle on the ANS in healthy subjects. An increase in parasympathetic and sympathetic activity was found via changes in heart rate variability after the acupuncture stimulation with an increase in LF (low frequency) both during and after the intervention. The results of this study support the postulate that dry puncturing has a simultaneous influence on the SNS and pain processing. In another study, the application of DN resulted in changes in the ANS, with subjects experiencing an increase in blood flow, both in skin and the muscle [51]. In contrast, another study evaluating the effects of DN on skin sympathetic activity found no significant changes after DN in a group of healthy subjects, but did find changes in skin sympathetic activity in the patient group [52]. In a recent study of healthy subjects in which acupuncture to tendons was applied, a change in local blood flow was observed, which was controlled by the SNS, but this was not related to heart rate [53]. In addition, when acupuncture was applied to trigger points in muscles such as the tibialis anterior, changes in heart rate were observed [54]. Contrary to the observations in our study, we are not aware of any previous studies in which puncturing was performed and an increase in heart rate was found. The study also suggests that the parasympathetic nervous system is involved in the relief mechanisms of myofascial pain through acupuncture stimulation.

In relation to mechanical hypoalgesia, other studies have shown an improvement in the PPT, and therefore are consistent with the results obtained in this work [28,29,55,56]. In contrast to these studies, it was concluded in a recent meta-analysis that DN shows low to moderate evidence of greater effects versus a placebo or control group in terms of improvement of pain and the pressure pain threshold [1].

However, these results contradict those previously found in other conducted studies, in which mechanical hyperalgesia occurs immediately after the DN is applied to healthy subjects, lasting up to 48 h [57,58,59]. It is possible that different pain processing mechanisms are activated in patients when DN is applied to active MTrPs, as shown by immediate increases found in previous studies. Dry needling performed in patients with musculoskeletal pain was found to improve mechanical hyperalgesia after applying different doses [28,29,60,61]. In another study, in which only a single DN session was applied, a percentage change of 54.85% was observed in PPT, with a difference of more than 5 percentage points between pre- and immediately post-treatment [62]. The mechanisms that explain these beneficial effects may occur due to an elimination of pronociceptive substances when DN is performed [63,64]. Alternatively, delta Aδ are stimulated, which activate the descending pain inhibitory systems as a counter-irritation mechanism [9,65]. Finally, the remote hypoalgesic effects obtained in this study contradict those observed by Sterling et al. [61] in patients with whiplash, in which local mechanical hyperalgesia improves, but this effect does not occur over a greater distance.

In a crossover clinical trial in which a dry needling-like therapy such as acupuncture was applied to whiplash patients, no relationship was found between pain improvement and changes in the SNS [36]. These results support those obtained in our study in the real DN group, but not in the placebo group, in which a positive correlation existed between the improvement in mechanical hyperalgesia at remote sites and the increment in the skin temperature. In the earlier research, the authors found that acupuncture produced a slight decrease in heart rate and an increase in skin conductance [36].

Importantly, salivary cortisol must be noted because it is considered to be the major indicator of stressful stimuli [17]. Cortisol works via circadian rhythms, reaching its maximum level approximately 30 min after waking. In our sample, we observed that the waking time of subjects in both groups was homogeneous. By comparison, the salivary cortisol values found in the literature that correspond to healthy adults in the same time slot as the measurements taken in this study are 0.43 μg/dL [17], which is similar to the basal values obtained in our sample.

In this study, no significant differences were found between the groups, although in the DN group a significant increase of 11.27% was found. The fact that no statistically significant differences were found between the two groups may be partly because saliva samples were not collected at the most opportune time. In the existing literature, no studies were found that observed the behavior over time of cortisol after a needle stimulus. Based on the collection of saliva samples in a study by Takai et al. [24], it was observed that after applying a stressful stimulus to psychologically healthy subjects, the highest salivary cortisol levels were obtained 3 min after the intervention.

The results obtained in the current research are supported by the study of Knardahl et al. [66], in which they measured plasma cortisol after electroacupuncture in healthy subjects, observing an immediate increase in cortisol in the intervention group and a decrease in the placebo group. No studies that measure salivary cortisol after applying DN were found in the literature; thus, more studies are needed to investigate the effect of this variable.

By comparison, several studies have described the fear of medical procedures, such as injections or dental care, as factors that produce fear and pain [67]. In the literature, studies have researched how pain induction produces changes in the ANS [68,69]. In the present study, the pain perception differed during the needling of both groups, and no relationship between the perceived pain during needling, and either the response of the ANS or changes in the PPT, was found. Therefore, pain might not be responsible for the changes observed in the subjects, and the results may be attributed to the effect of the technique.

Acupuncture is a procedure that consists of penetrating the skin with a needle, which can stimulate the primary nociceptor and induce pain. Lee et al. [70] performed a study to investigate the effects of acupuncture stimulation on ANS and its relationship with the fear of acupuncture. They found that skin conductance significantly increased after acupuncture stimulation and the fear of acupuncture-induced pain was associated with an enhanced physiological response. These data are consistent with those obtained in our study, in which we found a positive correlation between temperature changes and the state of anxiety, and between the heart rate and the level of catastrophism, in the group that received the real dry needling.

### 4.1. Clinical Implications

Dry needling is a technique used in the management of musculoskeletal pain; however, its specific mechanisms for modulating pain remain unknown. This study examined healthy subjects who were free of depression, anxiety, and pain catastrophism, and is the first step in understanding the effects of dry needling on the autonomic nervous system and nociceptive processing. The results of this study show a short-term improvement in PPT, not only locally, but also remotely, showing that dry needling does not produce changes only in the tissue, but also involves changes at the central level. Furthermore, it does not appear that the pain perceived by the subjects is the trigger for the changes produced at the physiological level. Finally, further studies are needed on the behavior of cortisol following needle stimulation. On the basis of these findings, further studies of subjects experiencing pain, with longer follow up, are needed to further investigate the effects of this technique at the central level.

### 4.2. Study Limitations

One limitation of this study is that the subjects may have previously experienced a DN treatment, and thus may have prior expectations about the intervention. In addition, the subjects were aware of the invasive treatment using a needle, and anticipation of the treatment may have influenced SNA activity. In future studies, a control group should be included to control for this fact. Moreover, the subjects were not aware of the existence of the placebo group, and whether the subjects had identified the group to which they belonged after the treatment was not monitored. In future studies, it will be necessary to include a blinding index to ensure the success of blinding.

The menstrual cycle in women should be monitored because of the influence it could have on the results.

Another possible limitation is the lack of close control of the state of wakefulness and sleep of the subjects, because this factor directly influences the responses of the SNS. Finally, only healthy subjects who were not experiencing pain participated in this study; thus, additional studies are necessary to evaluate the response of the ANS in acute and chronic pain processes, considering that the processes that modulate pain are different.

## 5. Conclusions

The results of this work showed that dry needling applied in healthy subjects immediately produced an increase in heart rate and a decrease in mechanical hyperalgesia, both locally and at remote sites, greater than that of the placebo intervention. Although the skin conductance, temperature, breathing rate, and cortisol levels also increased, no difference was found between the needling group and the placebo group. These results appear to indicate that dry needling produces an immediate activation in the sympathetic nervous system that are related to stress-induced analgesia mechanisms. Further studies are needed to clarify the possible implication of these underlying specific mechanisms.

## Figures and Tables

**Figure 1 ijerph-18-06018-f001:**
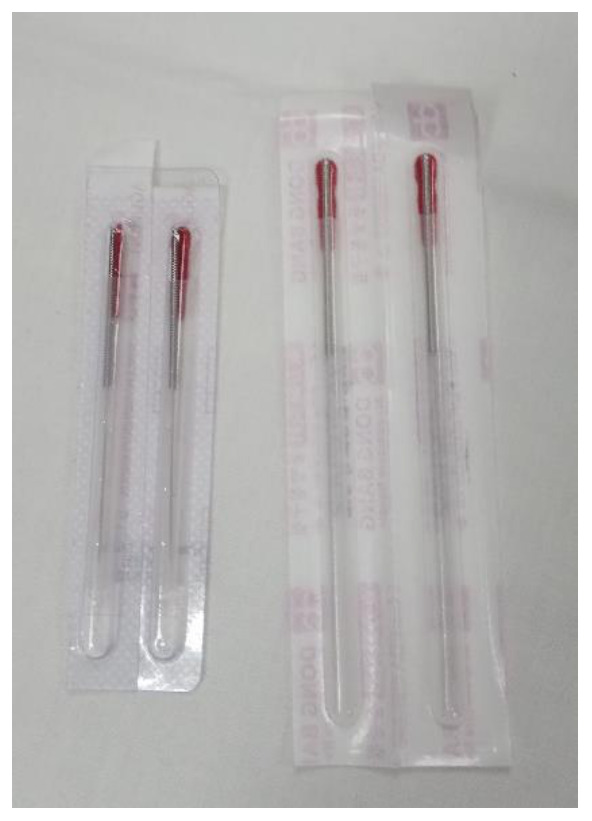
The needles used with the Deep Dry Needling Group are shown on the left; the needles used with the Placebo Needling Group are shown on the right.

**Figure 2 ijerph-18-06018-f002:**
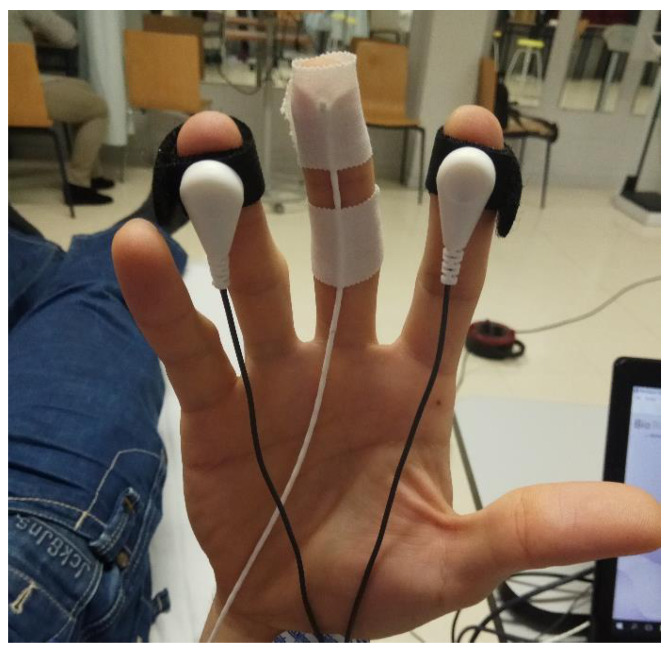
Placement of electrodes for SC and Temp on the right hand.

**Figure 3 ijerph-18-06018-f003:**
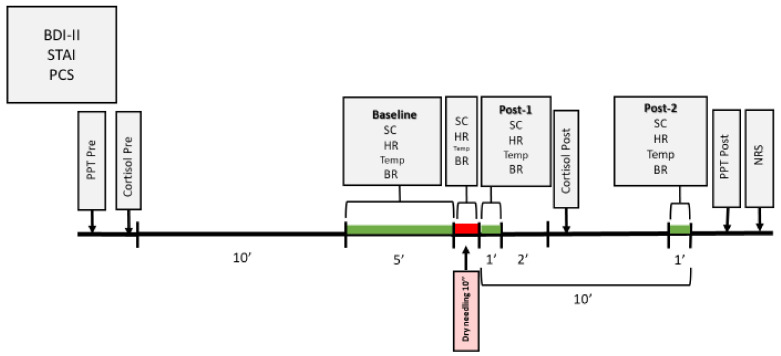
Temporal chronogram of the study.

**Figure 4 ijerph-18-06018-f004:**
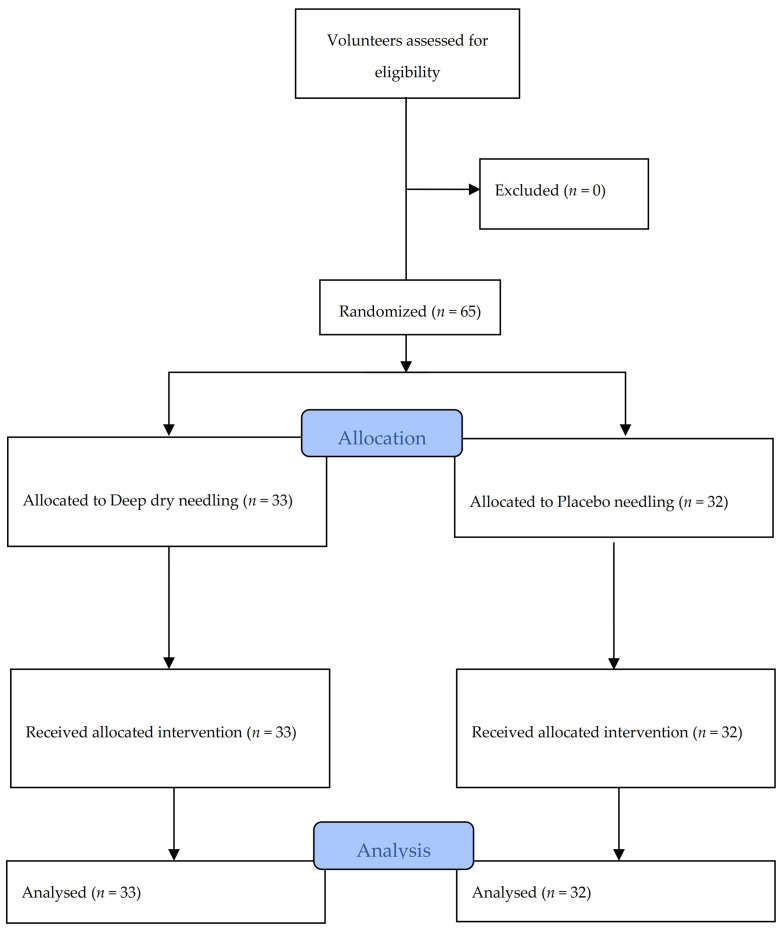
Flow diagram of patients throughout the course of the study.

**Table 1 ijerph-18-06018-t001:** Demographic characteristics of each treatment group.

Characteristics		Dry Needling Group (*n* = 33) Mean ± SD (Range)	Placebo Group (*n* = 32) Mean ± SD (Range)	*p*-Value *
Age (years)		26.88 ± 8.05 (19–51)	28.72 ± 8.79 (19–53)	0.382
Gender ^†^	Male	16 (48.5)	17 (53.1)	0.708 ^‡^
	Female	17 (51.5)	15 (46.9)	
Weight (kg)		71.28 ± 12.02 (46.6–98.4)	68.08 ± 16.00 (47.1–108.1)	0.366
Height (m)		1.71 ± 0.06 (1.58–1.82)	1.69 ± 0.08 (1.54–1.83)	0.262
BMI (kg/m^2^)		24.29 ± 3.52 (18.67–32.16)	23.66 ± 4.33 (17.78–33.36)	0.522
Wake-up time		7:28 ± 55.8 min (5:40–9:00)	7:36 ± 52.4 min (4:30–9:00)	0.565
BDI-II		5.85 ± 4.75 (0–21)	5.03 ± 4.04 (0–13)	0.458
STAI	State Anxiety	9.82 ± 7.48 (0–32)	12.34 ± 6.79 (0–29)	0.160
	Trait Anxiety	12.27 ± 7.47 (0–25)	13.31 ± 6.87 (0–25)	0.561
PCS	Rumination	2.97 ± 2.21 (0–8)	2.09 ± 2.94 (0–9)	0.179
	Helplessness	2.61 ± 2.79 (0–10)	1.66 ± 2.44 (0–9)	0.150
	Magnification	1.94 ± 1.87 (0–6)	1.31 ±1.65 (0–7)	0.158
	Total Score	7.00 ± 6.17 (0–21)	5.03 ± 5.69 (0–20)	0.187

* Comparison needling group versus placebo group using Student’s *t*-test. ^†^ Absolute frequency and category percentage *n* (%) are shown. ^‡^ Pearson’s chi-squared test was used. BMI: Body Mass Index; BDI-II: Beck Depression Inventory II; STAI: State-Trait Anxiety Inventory; PCS: Pain Catastrophizing Scale.

**Table 2 ijerph-18-06018-t002:** Primary and secondary outcomes.

Variables		Measurement	Dry Needling Group (*n* = 33)	Placebo Group (*n* = 32)	*p*-Value *	Effect Size Cohen’s d
SC (µs)		Baseline	2.95 ± 1.57	2.33 ± 1.26	0.082	0.43
		Needling	8.42 ± 4.06	7.71 ± 3.22	0.437	0.19
		Post-1	7.43 ± 3.43	6.01 ± 2.54	0.063	0.47
		Post-2	4.76 ± 2.49	4.12 ± 2.32	0.288	0.27
HR		Baseline	65.76 ± 11.29	68.39 ± 10.30	0.329	0.24
		Needling	78.93 ± 14.61	72.40 ± 16.51	0.096	0.42
		Post-1	66.50 ± 11.46	65.69 ± 12.04	0.784	0.07
		Post-2	62.96 ± 10.93	65.42 ± 10.08	0.349	0.23
Temperature (°C)		Baseline	30.81 ± 4.12	31.78 ± 3.79	0.275	0.24
		Needling	31.21 ± 3.88	31.72 ± 3.28	0.328	0.14
		Post-1	31.13 ± 3.82	31.64 ± 3.29	0.572	0.14
		Post-2	31.51 ± 3.93	32.61 ± 3.17	0.567	0.31
BR		Baseline	15.61 ± 3.48	14.03 ± 4.10	0.099	0.42
		Needling	21.26 ± 4.18	19.44 ± 4.01	0.079	0.44
		Post-1	19.24 ± 3.61	18.27 ± 4.29	0.326	0.25
		Post-2	16.52 ± 3.33	15.76 ± 5.14	0.478	0.18
Cortisol (µg/dL)		Baseline	0.53 ± 0.29	0.55 ± 0.27	0.734	0.07
		Post-test	0.59 ± 0.33	0.57 ± 0.32	0.822	0.06
PPT (Kg/cm^2^)	Thumb adductor	Baseline	2.13 ± 0.35	2.18 ± 0.43	0.562	0.13
		Post-test	2.72 ± 0.39	2.37 ± 0.43	0.001	0.85
	Anterior Tibialis	Baseline	4.96 ± 0.59	5.04 ± 0.64	0.630	0.13
		Post-test	5.57 ± 0.56	5.23 ± 0.62	0.022	0.58
NRS Pain		Post-test	5.87 ± 2.01	1.09 ± 1.17	0.001	2.89

Mean ± standard error (SE) * Comparison using Student’s *t*-test for independent samples. Bonferroni type adjustment was used. SC: Skin conductance; HR: Heart rate; BR: Breathing rate; PPT: Pressure pain threshold; NRS: Numeric Rating Scale for Pain.

**Table 3 ijerph-18-06018-t003:** % Change in primary outcomes and pressure pain threshold.

Variables	% Change between Measurements	Dry Needling Group (*n* = 33)	Placebo Group (*n* = 32)	*p*-Value *	Effect Size Cohen’s d
SC	Baseline to Needling	248.89 ± 37.66	323.14 ± 47.48	0.224	0.30
	Baseline to Post-1	214.45 ± 40.51	225.74 ± 36.02	0.836	0.05
	Baseline to Post-2	72.13 ± 10.54	90.91 ± 15.17	0.311	0.25
	Needling to Post-1	−10.36 ± 1.96	−21.50 ± 1.99	0.001	0.98
	Needling to Post-2	−41.78 ± 3.79	−45.39 ± 4.03	0.518	0.16
	Post-1 to Post-2	−35.16 ± 3.86	−31.28 ± 4.51	0.516	0.16
HR	Baseline to Needling	20.60 ± 2.88	5.33 ± 2.32	0.001	1.02
	Baseline to Post-1	1.40 ± 1.52	−4.10 ± 1.17	0.006	0.71
	Baseline to Post-2	−4.07 ± 1.05	−4.20 ± 0.90	0.927	0.02
	Needling to Post-1	−15.10 ± 1.47	−8.00 ± 1.67	0.002	0.79
	Needling to Post-2	−19.15 ± 1.92	−7.81 ± 1.98	0.001	1.02
	Post-1 to Post-2	−4.87 ± 8.39	0.16 ± 1.06	0.007	0.69
Temperature	Baseline to Needling	1.51 ± 0.78	0.14 ± 0.98	0.275	0.27
	Baseline to Post-1	1.27 ± 0.78	−0.11 ± 0.99	0.280	0.27
	Baseline to Post-2	2.59 ± 1.25	3.09 ± 1.20	0.773	0.07
	Needling to Post-1	−0.23 ± 0.13	−0.25 ± 0.16	0.961	0.02
	Needling to Post-2	1.01 ± 0.72	2.96 ± 0.69	0.055	0.49
	Post-1 to Post-2	1.25 ± 0.70	3.21 ± 0.65	0.045	0.51
BR	Baseline to Needling	43.13 ± 8.04	47.05 ± 7.66	0.725	0.09
	Baseline to Post-1	29.32 ± 7.08	36.82 ± 7.08	0.457	0.19
	Baseline to Post-2	9.24 ± 5.04	15.04 ± 5.46	0.438	0.19
	Needling to Post-1	−7.98 ± 2.77	−4.70 ± 3.63	0.473	0.18
	Needling to Post-2	−20.56 ± 3.04	−18.47 ± 3.87	0.671	0.11
	Post-1 to Post-2	−12.03 ± 3.47	−12.61 ± 3.94	0.912	0.03
Cortisol	Baseline to Post-test	11.27 ± 4.76	1.51 ± 3.08	0.920	0.42
PPT	Thumb adductor	28.57 ± 2.16	9.09 ± 1.40	0.001 ^ † ^	1.87
	Anterior Tibialis	12.69 ± 1.19	3.94 ± 0.59	0.001 ^ † ^	1.61

Mean ± standard error (SE). * Comparison using Student’s *t*-test for independent samples. ^†^ Welch’s *t*-test was used. Bonferroni type adjustment was used. SC: skin conductance; HR: heart rate; BR: breathing rate; PPT: pressure pain threshold.

**Table 4 ijerph-18-06018-t004:** Correlations of Numeric Rating Scale for Pain with % Change of the Physiological variables and Cortisol.

% Change between Measurements	NRS Dry Needling Group (*n* = 33)		NRS Placebo Group (*n* = 32)	
	Pearson’s Correlation Coefficient (r)	*p*-Value	Pearson’s Correlation Coefficient (r)	*p*-Value
SC Baseline—Needling	0.183	0.307	0.011	0.951
HR Baseline—Needling	0.049	0.786	0.039	0.833
Temperature Baseline—Needling	−0.227	0.204	0.033	0.857
BR Baseline—Needling	−0.015	0.933	−0.127	0.488
Cortisol Baseline—Post-test	0.102	0.572	0.117	0.523

Correlations using Pearson’s coefficient. SC: skin conductance; HR: heart rate; BR: breathing rate; NRS: Numeric Rating Scale for Pain.

**Table 5 ijerph-18-06018-t005:** Correlations between % Change of the Physiological Variables and Cortisol with Psychological Factors in the Dry Needling Group (*n* = 33).

Variables	% Change between Measurements	BDI-II	State Anxiety	Trait Anxiety	PCS Total
SC	Baseline to Needling	r = 0.024	r = 0.037	r = −0.016	r = −0.129
		*p* = 0.895	*p* = 0.840	*p* = 0.929	*p* = 0.473
	Needling to Post-1	r = 0.096	r = 0.093	r = 0.291	r = −0.122
		*p* = 0.596	*p* = 0.606	*p* = 0.100	*p* = 0.500
	Post-1 to Post-2	r = −0.044	r = 0.039	r = 0.202	r = 0.126
		*p* = 0.806	*p* = 0.828	*p* = 0.259	*p* = 0.484
HR	Baseline to Needling	r = 0.016	r = −0.070	r = −0.142	r = −0.079
		*p* = 0.929	*p* = 0.700	*p* = 0.429	*p* = 0.662
	Needling to Post-1	r = −0.219	r = −0.006	r = 0.052	r = −0.047
		*p* = 0.220	*p* = 0.974	*p* = 0.774	*p* = 0.797
	Post-1 to Post-2	r = 0.048	r = 0.077	r = 0.072	r = 0.081
		*p* = 0.790	*p* = 0.670	*p* = 0.689	*p* = 0.654
Temperature	Baseline to Needling	r = 0.007	r = −0.060	r = 0.041	r = −0.134
		*p* = 0.970	*p* = 0.738	*p* = 0.822	*p* = 0.458
	Needling to Post-1	r = 0.091	r = 0.350	r = 0.290	r = 0.202
		*p* = 0.614	*p* = 0.046	*p* = 0.101	*p* = 0.259
	Post-1 to Post-2	r = 0.058	r = −0.080	r = 0.143	r = −0.130
		*p* = 0.748	*p* = 0.659	*p* = 0.427	*p* = 0.471
BR	Baseline to Needling	r = 0.002	r = 0.023	r = 0.014	r = −0.291
		*p* = 0.990	*p* = 0.899	*p* = 0.940	*p* = 0.101
	Needling to Post-1	r = −0.006	r = −0.203	r = −0.003	r = −0.126
		*p* = 0.974	*p* = 0.258	*p* = 0.986	*p* = 0.486
	Post-1 to Post-2	r = −0.035	r = 0.061	r = −0.85	r = 0.393
		*p* = 0.845	*p* = 0.735	*p* = 0.637	*p* = 0.024
Cortisol	Baseline to Post-test	r = 0.043	r = 0.211	r = −0.067	r = 0.042
		*p* = 0.813	*p* = 0.239	*p* = 0.713	*p* = 0.815

Correlations using Pearson’s coefficient (r) and *p*-value (*p*) SC: skin conductance; HR: heart rate; BR: breathing rate; BDI-II: Beck Depression Inventory II; PCS: Pain Catastrophizing Scale.

**Table 6 ijerph-18-06018-t006:** Correlations between % Change of the Physiological Variables and Cortisol with Psychological Factors in the Placebo Group (*n* = 32).

Variables	% Change between Measurements	BDI-II	State Anxiety	Trait Anxiety	PCS Total
SC	Baseline to Needling	r = 0.157	r = 0.013	r = −0.041	r = 0.001
		*p* = 0.392	*p* = 0.943	*p* = 0.823	*p* = 0.999
	Needling to Post-1	r = −0.055	r = 0.072	r = 0.045	r = 0.078
		*p* = 0.767	*p* = 0.695	*p* = 0.807	*p* = 0.673
	Post-1 to Post-2	r = −0.153	r = −0.196	r = −0.093	r = 0.179
		*p* = 0.403	*p* = 0.283	*p* = 0.612	*p* = 0.327
HR	Baseline to Needling	r = 0.111	r = −0.039	r = 0.070	r = 0.009
		*p* = 0.544	*p* = 0.834	*p* = 0.705	*p* = 0.961
	Needling to Post-1	r = −0.120	r = 0.074	r = 0.035	r = 0.050
		*p* = 0.514	*p* = 0.689	*p* = 0.851	*p* = 0.786
	Post-1 to Post-2	r = −0.086	r = −0.254	r = −0.196	r = −0.36
		*p* = 0.639	*p* = 0.161	*p* = 0.282	*p* = 0.846
Temperature	Baseline to Needling	r = −0.121	r = 0.053	r = 0.024	r = 0.222
		*p* = 0.510	*p* = 0.775	*p* = 0.895	*p* = 0.222
	Needling to Post-1	r = 0.111	r = 0.056	r = 0.160	r = −0.076
		*p* = 0.544	*p* = 0.761	*p* = 0.381	*p* = 0.678
	Post-1 to Post-2	r = 0.061	r = 0.016	r = −0.119	r = −0.025
		*p* = 0.741	*p* = 0.932	*p* = 0.517	*p* = 0.890
BR	Baseline to Needling	r = 0.042	r = 0.098	r = 0.211	r = −0.159
		*p* = 0.819	*p* = 0.593	*p* = 0.247	*p* = 0.385
	Needling to Post-1	r = −0.139	r = 0.138	r = 0.053	r = 0.155
		*p* = 0.447	*p* = 0.453	*p* = 0.773	*p* = 0.398
	Post-1 to Post-2	r = −0.018	r = −0.354	r = −0.325	r = 0.002
		*p* = 0.923	*p* = 0.047	*p* = 0.069	*p* = 0.993
Cortisol	Baseline to Post-test	r = −0.407	r = −0.416	r = −0.371	r = 0.064
		*p* = 0.021	*p* = 0.018	*p* = 0.037	*p* = 0.728

Correlations using Pearson’s coefficient (r) and *p*-value (*p*) SC: skin conductance; HR: heart rate; BR: breathing rate; BDI-II: Beck Depression Inventory II; PCS: Pain Catastrophizing Scale.

## Data Availability

The data presented in this study are available on request from the corresponding author. The data are not publicly available due to ethical condition.

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
