# Peer review of "Immediate Effects of Dry Needling on the Autonomic Nervous System and Mechanical Hyperalgesia: A Randomized Controlled Trial"

_ijerph, 2021, doi:10.3390/ijerph18116018_

Round 1
Reviewer 1 Report
Generally speaking, the paper is well-structured and well-written. The research findings are clearly presented. However, the use of English language needs improvements. There are quite many grammatical errors in the paper e.g.
- What does "it" in "in order to use it" in line 209 stands for?
- Hyphen is not needed in the word "exam-ine" in line 253.
- It should be "moderates" rather than "modulates" in line 464.
The authors should have the paper proof-edited by professional English writer before submission.
Limitations have been adequately acknowledged in the paper. However, I just wonder if some participants in the control knew that they were not treated with Deep Dry Needle? It is likely that some people did have previous experience of Deep Dry Needle before so these people could know they were in placebo group.
Besides, the paper has some issues of plagiarism. The authors should be reminded that even though they have acknowledged the sources, they should rephrase the wordings from the original sources (unless they are doing direct quotations). As shown in the attached iThenticate report, direct copying is quite often in Sections 1 and 2. The paper should be overhauled to clear the plagiarism issue.

Author Response
We would like to thank the reviewers for the insightful comments, which have greatly enhanced the contribution of our paper and have improved the readability of the manuscript. Below please find a list of revisions and a response to each of the reviewers’ comments. We have highlighted all changes in the manuscript and referred to the specific reviewer query in the text
Reviewer 1
Generally speaking, the paper is well-structured and well-written. The research findings are clearly presented. However, the use of English language needs improvements. There are quite many grammatical errors in the paper e.g.
- What does "it" in "in order to use it" in line 209 stands for?
Response:
The phrase has been modified for a better understanding in line 214: “To implement this scale”
- Hyphen is not needed in the word "exam-ine" in line 253:
Response:
The hyphen has been eliminated. In line 260 it has also been found a hyphen in the word “variable”: the hyphen has been also eliminated.
- It should be "moderates" rather than "modulates" in line 464:
Response:
Thank you very much for the suggestion. However, in the way the research team understands it, the word “modulates” suits with the meaning of the sentence.
The authors should have the paper proof-edited by professional English writer before submission.
Response: the article has already been sent to the English grammar edition and the invoice has been sent. The article has been water-proof-edited by a professional English writer, and the proposed changes have been introduced in the manuscript.
Limitations have been adequately acknowledged in the paper. However, I just wonder if some participants in the control knew that they were not treated with Deep Dry Needle? It is likely that some people did have previous experience of Deep Dry Needle before so these people could know they were in placebo group.
Response: As the reviewer has pointed out subjects were not asked about their precognitions on their allocation to either the intervention or the control groups. For this reason, unfortunately, we are unable to get an answer to this question. This was included as a study limitation, as well as the fact that subjects could have previously experienced a previous DN treatment.
When methodological procedures of this research were designed, the members of the research team followed the recommendations displayed in previous investigations, in order to implement an effective placebo technique.1,2 In these studies, they did not consider the exclusion of subjects with prior experience on invasive techniques: “Patients with previous experience of acupuncture were not excluded for two reasons. Firstly because if a placebo intervention is to have true validity, it must be able to be used with both naïve and experienced patients, and secondly because there is as yet no evidence to suggest that this needle should be confined to acupuncture naïve patients only.”1
Nevertheless, in order to minimize methodological bias we excluded those participants that had received a dry needling/acupuncture treatment within the previous 6 months, similarly to other comparable studies3. On the other hand, to ensure that both groups were comparable, before the start of the study, the subjects were asked if they had previously received dry needling treatment. The results are shown in the following table, in which it can be seen that there are no differences between groups.
Previous experience of Deep Dry Needle |
|
Deep Dry Needling |
Placebo |
P-VALUE |
YES |
19 (57.6%) |
19 (59.4%) |
P= 0,883* |
|
NOT |
14 (42.4%) |
13 (40.6%) |
* Pearson Chi-Squared
However, a recent systematic review from 20194, which was published after our study was completed, confirmed the importance of recruiting subjects with no prior needle treatments.4 Although we could not take this findings into consideration, we followed the rest of the recommendations proposed by Braithwaite et al.4 and fulfilled them in our study.
If you consider it important to enter these dates to the manuscript, please let us know.
- White P, Lewith G, Hopwood V, Prescott P. The placebo needle, is it a valid and convincing placebo for use in acupuncture trials? A randomised, single-blind, cross-over pilot trial. Pain. 2003;106(3):401–9.
- Tough EA, White AR, Richards SH, Lord B, Campbell JL. Developing and validating a sham acupuncture needle. Acupunct Med. 2009;27(3):118–22.
- Bäcker M, Grossman P, Schneider J, Michalsen A, Knoblauch N, Tan L, et al. Acupuncture in migraine: investigation of autonomic effects. Clin J Pain. 2008 Feb;24(2):106-15. doi: 10.1097/AJP.0b013e318159f95e.
- Braithwaite FA, Walters JL, Li LSK, Moseley GL, Williams MT, McEvoy MP. Blinding Strategies in Dry Needling Trials: Systematic Review and Meta-Analysis. Phys Ther. 2019;99(11):1461–80.
Besides, the paper has some issues of plagiarism. The authors should be reminded that even though they have acknowledged the sources, they should rephrase the wordings from the original sources (unless they are doing direct quotations). As shown in the attached iThenticate report, direct copying is quite often in Sections 1 and 2. The paper should be overhauled to clear the plagiarism issue
Response:
Thank you very much for the document attached and the review of the plagiarism. We have reviewed and rephrased some of the sentences following the suggested directions.
Reviewer 2 Report
Dear co-authors;
I congratulate you for the effort made in this study.
The manuscript is very well organized, and in a general way it is clear in the description of the study and methodology. However, I think there are different aspects that should be clarified. I think the title affirms something that cannot be affirmed, and the fact is that they do not affirm it in conclusions, so they must conform to the findings, even if that was their objective.
In the introduction, I would explain the different theories or approaches in research to other causes of pain reduction after dry needling, such as the activation of the descending pain inhibition mechanism at the CNS level.
Line 153- I think it is necessary to explain in the methodology if the subjects, being a student, know the puncture technique or had received it previously, because during the placebo application they could notice the difference although visually the placebo needle did not distinguish the feeling during procedure varies between placebo and dry needling.
Line 239- Explain why you use the independent samples t test for quantitative variables and the Pearson’s Chi-Squared test for qualitative variables, if the first is a parametric test and the second is a non-parametric test.
Line 240- should better explain why it normalized all the variables, especially when there are no differences between groups initially, was it really necessary in all the variables?
Line 256- State the reason for adding the statistic of the size of the effect found. All tables- It would be appropriate to add the statistic or calculation performed at the bottom of each table.
Line 495 would eliminate it if it cannot be affirmed, it should not be drawn to conclusions.
Author Response
We would like to thank the reviewers for the insightful comments, which have greatly enhanced the contribution of our paper and have improved the readability of the manuscript. Below please find a list of revisions and a response to each of the reviewers’ comments. We have highlighted all changes in the manuscript and referred to the specific reviewer query in the text.
I congratulate you for the effort made in this study.
The manuscript is very well organized, and in a general way it is clear in the description of the study and methodology. However, I think there are different aspects that should be clarified. I think the title affirms something that cannot be affirmed, and the fact is that they do not affirm it in conclusions, so they must conform to the findings, even if that was their objective.
In the introduction, I would explain the different theories or approaches in research to other causes of pain reduction after dry needling, such as the activation of the descending pain inhibition mechanism at the CNS level.
Response:
We have added this paragraph to the text (lines 56-62):
The possible explanations found in the literature for the decrease in pain include the effects of dry needling at the local level (producing an interruption of spontaneous electrical activity on the taut band or local vasodilation), activation of the peripheral segmental pain inhibition (explained through Gate Control Theory), or activation of the descending pathways of pain inhibition at the central nervous system level (serotonergic and noradrenergic endogenous opioid release and conditioned modulation of pain)
Line 153- I think it is necessary to explain in the methodology if the subjects, being a student, know the puncture technique or had received it previously, because during the placebo application they could notice the difference although visually the placebo needle did not distinguish the feeling during procedure varies between placebo and dry needling.
Response:
In study limitations, we highlighted he fact that the subjects could have previously experienced a DN treatment. On the other hand, the sample has been made up of students and administrators from the University of Alcalá. Some of the students were physiotherapists (although dry needling is not taught in the Physiotherapy Degree subjects), and nurses, but subjects from other faculties not related to health were also included. In the methodology we cannot think of how to include these details, but if you consider it important, we can include as limitations of the study that the previous knowledge that the subjects had about the dry needling technique was not recorded, since this knowledge about the technique can influence blinding correct technique.
Line 239- Explain why you use the independent samples t test for quantitative variables and the Pearson’s Chi-Squared test for qualitative variables, if the first is a parametric test and the second is a non-parametric test.
Response:
Non-parametric tests, such as the Pearson's Chi-Squared test are used to compare qualitative or categorical variables (as in this case is sex), and in the case of the rest of the variables studied, which are numerical (quantitative) variables that are also conform to normal, the independent samples t test has been used.
Line 240- should better explain why it normalized all the variables, especially when there are no differences between groups initially, was it really necessary in all the variables?
Response:
We consider interesting not only showing the values obtained in the different measurements but also analyzing the changes that occurred in the different periods. For this purpose, it describes in relation to the initial level how this data has behaved.
In addition to showing the values in the different measurements, as we do in Table 2, we have shown the percentages of change in order to calculate these differences, putting the initial starting value in context, in order to better compare those changes.
I show some bibliographic references in which the data are also analyzed as percentages of change, as in the article published in Manual Therapy1, in another article by Choi et al.2 in which they study the effects of acupuncture on the autonomic nervous system or the of Abbaszadeh-Amirdehi et al.3 in 2016.
- Perry, J.; Green, A. An investigation into the effects of a unilaterally applied lumbar mobilisation technique on peripheral sympathetic nervous system activity in the lower limbs. Ther. 2008, 13, 492–9, doi:10.1016/j.math.2007.05.015
- Choi W, Lee S, Cho S, Park K. Differential autonomic response to acupuncture at wood and metal of five-shu acupoints. J Altern Complement Med. 2012, 18, 959–64.
- Abbaszadeh-Amirdehi M, Ansari NN, Naghdi S, Olyaei G, Nourbakhsh MR. Therapeutic effects of dry needling in patients with upper trapezius myofascial trigger points. Acupunct Med. 2017, 35, 85-92,doi: 10.1136/acupmed-2016-011082.
Line 256- State the reason for adding the statistic of the size of the effect found. All tables- It would be appropriate to add the statistic or calculation performed at the bottom of each table.
Response:
The p-value informs us is whether there are differences between the groups if the value is significant, but it does not provide us with more information. For this reason, in this study, in addition to giving the p-value, we show the effect size coefficient, in order to quantify the magnitude of these differences found: if they are between 0.2 and 0.5, it informs us of a “small” effect size, between 0.5 and 0.5. 0.8 to "medium" effect size, and > 0.8 to "large" effect size. With this we are providing more information to the data shown, not only whether or not there are differences between the groups.
If you consider that it is not necessary to enter this statistic, please let us know and we will modify it in a later revision.
Following their recommendations, the corresponding statistics carried out have been added to each table at the bottom of each table.
Line 495 would eliminate it if it cannot be affirmed, it should not be drawn to conclusions.
Response:
This phrase is supported by the results shown in Table 3, which shows statistically significant differences are obtained in the increase of the heart rate between the two groups between the baseline and the needling (p = 0.001) with a large effect size (d = 1.02).
Reviewer 3 Report
This study suggested very interesting results and experimental suggestions.
Also, this paper is well written with logical flow.
However, I found some major limitations and suggest revision related these issues.
This paper deserves to be published after major revisions.
1. It was said that 65 subjects were divided into 33 dry needling group and 32 placebo group, and the grouping was randomized. Is it a coincidence that the male and female ratio was divided by one or two like that?
2. In figure 1., why is there a difference in the size of needles used between the two groups? Didn't the difference affect the results as well?
3. It was said that the subject was between the ages of 18 and 65, but is there any difference in the autonomic nervous system and pain sensation in the case of an elderly subject like a 53 year old subject?
Most of them are in their twenties, but how many people participated in the age difference?
Didn't that affect the research results?
4. The subjects were normal people, and the intervention was conducted. How long and how many times did you perform the intervention? Do you think it was a no-problem mediation even if it proceeded to a normal person?
5. Is there any moral problem in the areas where the subjects were not allowed to brush their teeth before participating in the study, nor did they eat or drink water before participating in the study?
6. Check the references. Read the instruction to authors in detail.
Author Response
We would like to thank the reviewers for the insightful comments, which have greatly enhanced the contribution of our paper and have improved the readability of the manuscript. Below please find a list of revisions and a response to each of the reviewers’ comments. We have highlighted all changes in the manuscript and referred to the specific reviewer query in the text.
This study suggested very interesting results and experimental suggestions.
Also, this paper is well written with logical flow.
However, I found some major limitations and suggest revision related these issues.
This paper deserves to be published after major revisions.
- It was said that 65 subjects were divided into 33 dry needling group and 32 placebo group, and the grouping was randomized. Is it a coincidence that the male and female ratio was divided by one or two like that?
Response:
The proportion found by group in this study by sex (Dry needling group: 48,5% male and 51,5% female; Placebo Group 53,1% male and 46,9% female) has not been manipulated or assigned in any way other than random. To carry out this study, each participant was previously assigned a code, and a person outside the study performed a concealed randomization using the Epidat 4.2 program.
- In figure 1., why is there a difference in the size of needles used between the two groups? Didn't the difference affect the results as well?
Response:
The needles of the respective groups are different: in the Deep dry needling group, was performed with disposable needles (0.25 x 0.25 mm; AGU-A1038P; Agu-Punt S.L, Barcelona, Spain); and in the Placebo Needling Group, placebo needles (0.25 x 0.40 mm; DB100-2540; DongBand, AcuPrime®, Exeter, UK) have been used. Placebo needles have been used following the recommendations to implement an effective placebo technique1,2, and the characteristic of these needles is that they have a spring handle that can be glided up and down. As shown in figure 1, at first glance the only difference between the two is the size of the needle, trying to make the rest of the characteristics as similar as possible. Also, the placebo needles were in the same box as the normal needles to avoid possible bias, and the subjects only saw their own needles, not those of the other subjects. Since the same diameter of the needle is present in both groups, we do not consider that the greater length of the needle may affect the results.
- White P, Lewith G, Hopwood V, Prescott P. The placebo needle, is it a valid and convincing placebo for use in acupuncture trials? A randomised, single-blind, cross-over pilot trial. Pain. 2003;106(3):401–9.
- Tough EA, White AR, Richards SH, Lord B, Campbell JL. Developing and validating a sham acupuncture needle. Acupunct Med. 2009;27(3):118–22.
It was said that the subject was between the ages of 18 and 65, but is there any difference in the autonomic nervous system and pain sensation in the case of an elderly subject like a 53 year old subject? Most of them are in their twenties, but how many people participated in the age difference? Didn't that affect the research results?
Response:
We have carried out analyzes to see if pain is related to the age of our subjects, and we have seen that pain is not related to age, neither when analyzing all the patients together nor when disaggregating them by groups.
On the other hand, when comparing the age in both groups with each other, it has been seen that there is no age difference, and therefore the data are comparable.
If we focus on the existing bibliography, to evaluate whether changes on the autonomic nervous system influence age, they usually set the age cut-off quite high: the study by Reardon and Malik1 studied whether age affects the autonomic nervous system. To do this, they study the subjects, dividing them by age groups of> or <to 70 years. In the study by Mohamed et al. speak of elderly people over 60-65 years2. In the studies consulted that evaluate the effects of invasive physiotherapy techniques, they also use an age range similar to that of our study3-5.
However, if it would be interesting to observe the results on pain and the autonomic nervous system in subjects disaggregated according to their age, although these objectives are beyond those of the present study.
- Reardon M, Malik M. Changes in heart rate variability withage. Pacing Clin Electrophysiol 1996. https://doi.org/10.1111/j.1540-8159.1996.tb03241.x
- Mohamed Nabil Alama, Aging-Related Changes of the Cardiovascular System, Journal of Health and Environmental Research. Vol. 3, No. 2, 2017, pp. 27-30. doi: 10.11648/j.jher.20170302.12
- Benito-de-Pedro, Becerro-de-Bengoa-Vallejo, Losa-Iglesias, Rodríguez-Sanz, López-López, Cosín-Matamoros, et al. Effectiveness between Dry Needling and Ischemic Compression in the Triceps Surae Latent Myofascial Trigger Points of Triathletes on Pressure Pain Threshold and Thermography: A Single Blinded Randomized Clinical Trial. J Clin Med. 2019;8(10):1632.
- Sillevis R, Van Duijn J, Shamus E, Hard M. Time effect for in-situ dry needling on the autonomic nervous system, a pilot study. Physiother Theory Pract. 2019;17:1–9.
- Abbaszadeh-Amirdehi M, Ansari NN, Naghdi S, Olyaei G, Nourbakhsh MR. Neurophysiological and clinical effects of dry needling in patients with upper trapezius myofascial trigger points. J Bodyw Mov Ther. 2017,21:48–52.
- The subjects were normal people, and the intervention was conducted. How long and how many times did you perform the intervention? Do you think it was a no-problem mediation even if it proceeded to a normal person?
Response:
All the subjects received only one session of deep dry needling or the placebo needling. I have included it in line 128 of the manuscript to emphasize it: “All of the subjects received one session of deep dry needling or placebo needling”. As mentioned in the line 151, the needle was moved up and down at 1Hz frequency for 10 seconds in the group Deep Dry Needling; and as mentioned in the line 167, in the Placebo Group: 10 times at a speed of 1Hz.
The dry needling technique is widely used in clinical practice. To the best of our knowledge, we do not believe that there may be any complication to be able to apply this technique to observe the effects on pain and the autonomic nervous system in healthy subjects. In the existing bibliography, there are studies in which they have followed the same methodology of both dry needling1,2 and acupuncture3-5, as in other studies they have applied stimuli of another nature to observe the behavior of the autonomic nervous system6.
- Sillevis R, Van Duijn J, Shamus E, Hard M. Time effect for in-situ dry needling on the autonomic nervous system, a pilot study. Physiother Theory Pract. 2019;17:1–9
- Benito-de-Pedro, Becerro-de-Bengoa-Vallejo, Losa-Iglesias, Rodríguez-Sanz, López-López, Cosín-Matamoros, et al. Effectiveness between Dry Needling and Ischemic Compression in the Triceps Surae Latent Myofascial Trigger Points of Triathletes on Pressure Pain Threshold and Thermography: A Single Blinded Randomized Clinical Trial. J Clin Med. 2019;8(10):1632.
- Kang OS, Chang DS, Lee MH, Lee H, Park HJ, Chae Y. Autonomic and subjective responses to real and sham acupuncture stimulation. Auton Neurosci Basic Clin 2011, 159:127–30.
- Paulson KL, Shay BL. Sympathetic nervous system responses to acupuncture and non-penetrating sham acupuncture in experimental forearm pain: a single-blind randomised descriptive study. Acupunct Med 2013, 31:178–84.
- Bäcker M, Schaefer F, Siegler N, Balzer S, Michalsen A, Langhorst J, et al. Impact of stimulation dose and personality on autonomic and psychological effects induced by acupuncture. Auton Neurosci Basic Clin. 2012, 170:48–55.
- Takai, N.; Yamaguchi, M.; Aragaki, T.; Eto, K.; Uchihashi, K.; Nishikawa, Y. Effect of psychological stress on the salivary cortisol and amylase levels in healthy young adults. Oral Biol. 2004, 49, 963–968, doi:10.1016/j.archoralbio.2004.06.007.
Is there any moral problem in the areas where the subjects were not allowed to brush their teeth before participating in the study, nor did they eat or drink water before participating in the study?
Response:
Through the patient information sheet, before the subjects voluntarily decided to participate in the study, it was reported among others, that they were not allowed to brush their teeth, eat solid food, drink any liquid or chew gum within 30 minutes before the study. Prior to that half hour they did not have those restrictions. The reason for these guidelines was to follow the recommendations of the Cortisol ELISA® kit from IBL International Laboratory (Hamburg, Germany) to collect saliva samples, and so that cortisol levels in saliva can be correctly analyzed. Also these guidelines have been given in the bibliography to take this type of samples1,2: the fact of not brush the teeth for avoid micro-vascular leakage into the sample, and the avoid eat, drink due to the post-prandial increases in adrenocortical activity.
On the other hand, prior to the start of the study, the methodology of this study was approved by the Ethics Committee of the University of Alcalá (CEIT/HU/2015/06). For these reasons we do not consider that there is any moral problem.
- Huang, W.; Taylor, A.; Howie, J.; Robinson, N. Is the Diurnal Profile of Salivary Cortisol Concentration a Useful Marker for Measuring Reported Stress in Acupuncture Research? A Randomized Controlled Pilot Study. J. Altern. Complement. Med. 2012, 18, 242–250, doi:10.1089/acm.2010.0325.
- Edwards, S.; Evans, P.; Hucklebridge, F.; Clow, A. Association between time of awakening and diurnal cortisol secretory activity. Psychoneuroendocrinology 2001, 26, 613–622, doi:10.1016/S0306-4530(01)00015-4.
- Check the references. Read the instruction to authors in detail:
Response:
The references have been modified and adapted according to the regulations of the journal. Changes marked "Track Changes" are displayed.
Reviewer 4 Report
- IRB approval is necessary, please verify IRB approval # (Subjects are vulnerable group.
- Blinding Index need to be included to ensure the results.
Author Response
We would like to thank the reviewers for the insightful comments, which have greatly enhanced the contribution of our paper and have improved the readability of the manuscript. Below please find a list of revisions and a response to each of the reviewers’ comments. We have highlighted all changes in the manuscript and referred to the specific reviewer query in the text.
- IRB approval is necessary, please verify IRB approval # (Subjects are vulnerable group.
Response:
To the best of our knowledge, an institutional review board (IRB) is an ethics committee of the United States, establishing a constituted group that has been formally designated to review and monitor biomedical research involving human subjects. The Committee of Research Ethics and Animal Experimentation of the University of Alcalá1, likewise, is an official entity that independently reviews studies carried out on human participants in Spain. This clinical trial was reviewed and approved by this institution on November 23, 2015 (CEIT/HU/2015/06), and to carry it out we strictly adhere to the protocol presented in it. The approval document has already been sent to the journal.
Similarly, the subjects who have participated in this study have been volunteers who were previously informed of the entire intervention protocol.
1 Comité de Ética de Investigación y Experimentación Animal de la Universidad de Alcalá. Available online: https://www.uah.es/es/investigacion/servicios-para-el-investigador/comite-de-etica-de-investigacion-y-experimentacion-animal/ (accessed on 16 April 2021)
- Blinding Index need to be included to ensure the results
Response:
Unfortunately, we are unable to obtain an answer to this question, as the subjects were not asked about their allocation to either the intervention or the control groups, and therefore we cannot calculate the blinding rate. We have included it in the limitations since we consider that for future research it is necessary to calculate this index (line 485): “In future studies, it will be necessary to include a blinding index to ensure the success of blinding”
Round 2
Reviewer 2 Report
Dear authors,
The new version made improves the previous one and for that I want to congratulate you.
However, and based on the statement they make in lines 400-401, the changes in the relationship to the autonomic nervous system are only observed in the heart rate, and the heart does not have only exclusively autonomic innervation.
- I understand that the authors known to other studies and dependent on the results of this study and that of other authors affirm that dry needling makes changes in the autonomous NS, but they were studies with longer applications (line 402-404)
Therefore, I would express that the immediate application of dry needling, as is done in this study, causes an increase in heart rate or activation of the sympathetic nervous system, but we have no evidence of what it states in the title that it affects the autonomic nervous system function.
- I would express it in the same way in the conclusions.
- It would also include in methods that the volunteers who participated in the study were students of various studies, not just physical therapy.
Author Response
Reviewer 2
Dear authors,
The new version made improves the previous one and for that I want to congratulate you.
However, and based on the statement they make in lines 400-401, the changes in the relationship to the autonomic nervous system are only observed in the heart rate, and the heart does not have only exclusively autonomic innervation.
- I understand that the authors known to other studies and dependent on the results of this study and that of other authors affirm that dry needling makes changes in the autonomous NS, but they were studies with longer applications (line 402-404)
Therefore, I would express that the immediate application of dry needling, as is done in this study, causes an increase in heart rate or activation of the sympathetic nervous system, but we have no evidence of what it states in the title that it affects the autonomic nervous system function.
- I would express it in the same way in the conclusions.
- It would also include in methods that the volunteers who participated in the study were students of various studies, not just physical therapy.
Response:
Thank you very much for your input and your suggestions.
Changes have been made to the abstract conclusions (line 37-38): “This work appears to indicate that dry needling produces an immediate activation in the sympathetic nervous system, improving local and distant mechanical hyperalgesia”
Changes have also been made to the conclusions of the manuscript (line 499-501): “These results appear to indicate that dry needling produces immediate changes in the sympathetic nervous system that are related to stress-induced analgesia mechanisms”
Regarding the title, a modification has been made:
“Immediate effects of dry needling on the autonomic nervous system and mechanical hyperalgesia: a randomized controlled trial”
Finally, we have added in the methodology that the subjects belonged to different degrees (line 102): “The sample comprised healthy volunteers from the student body of different degrees and the administrative staff of the University of Alcalá”

Reviewer 4 Report
Since the author made clear about the ethical problem, it could be accepted in present form.
Author Response
No requests for changes have been made by the reviewer 4